# Limitations of 18S rDNA Sequence in Species-Level Classification of Dictyostelids

**DOI:** 10.3390/microorganisms13020275

**Published:** 2025-01-26

**Authors:** Thanyaporn Chittavichai, Sukhita Sathitnaitham, Supanut Utthiya, Wanasilp Prompichai, Kamonchat Prommarit, Supachai Vuttipongchaikij, Passorn Wonnapinij

**Affiliations:** 1Department of Genetics, Faculty of Science, Kasetsart University, Bangkok 10900, Thailand; thanyaporn.chit@ku.th (T.C.); sukhita.sa@ku.th (S.S.); supanut.ut@ku.th (S.U.); wanasilp.p@gmail.com (W.P.); kamonchat.p@ku.th (K.P.); supachai.v@ku.th (S.V.); 2Centre for Advanced Studies in Tropical Natural Resources, Kasetsart University, Bangkok 10900, Thailand; 3Omics Center for Agriculture, Bioresources, Food and Health, Kasetsart University, Bangkok 10900, Thailand

**Keywords:** delimitation, dictyostelids, DNA marker, species, SSU

## Abstract

Dictyostelid species classification has traditionally relied on morphology, a time-intensive method requiring expert knowledge. This study evaluated the potential and limitations of using the 18S rDNA sequence for species-level classification. 18S rDNA sequences of 16 samples from the Dicty stock center, including 14 samples found in Thailand, were analyzed. Signature sequence analyses confirmed genus-level identification with high accuracy. These sequences were analyzed alongside 309 database entries retrieved from the GenBank database. The analyses confirmed genus-level identification accuracy but highlighted challenges in distinguishing species due to overlapping intraspecific and interspecific variations, negative barcoding gaps, and incorrectly grouped samples to putative taxa by species delimitation analyses. Species delimitation methods, including maximum likelihood (ML) phylogenetic analysis, achieved limited success, with ML showing the highest accuracy but not exceeding 50%. However, species with high barcoding gaps, such as *Raperostelium* and *Rostrostelium*, demonstrated potential for accurate classification. These findings support using 18S rDNA for genus-level identification and suggest its possible application for certain species. Expanded sampling is needed to improve species-level classification and to identify more robust DNA markers for dictyostelid diversity studies.

## 1. Introduction

Dictyostelids, known as social amoebae, are soil-dwelling eukaryotic microbes characterized by four distinct life cycles: vegetative, encystation, multicellular sexual, and multicellular sorocarpic cycles [1]. Under normal conditions, these organisms exist in the vegetative cycle as amoeboid cells that feed on bacteria and other microorganisms in the soil. During periods of food scarcity, dictyostelids can enter either the encystation cycle or one of the multicellular cycles. Encystation, a process where amoebae form protective cysts, has been observed in all dictyostelid genera except *Dictyostelium*. The multicellular sexual and sorocarpic cycles, on the other hand, are triggered primarily by starvation, though dark and humid conditions can also promote the sexual cycle [1].

In the multicellular sorocarpic cycle, dictyostelids release chemoattractants to initiate cell aggregation. These attractants are group- or species-specific; for instance, cyclic AMP (cAMP) induces aggregation in *Dictyostelium discoideum*, while glorin serves a similar role in *Polysphondylium violaceum* [2]. Following aggregation, the cells form mounds, sorogens, and sorocarps (fruiting bodies). In *Dictyostelium*, motile sorogens (slugs) are formed and culminate into sorocarps [1]. The sorocarp, or fruiting body, serves as a reproductive structure. While *Dictyostelium* exhibits a complex sorocarpic structure, other genera form simpler sorocarps consisting only of stalk and spore cells [3]. Morphologically, sorocarps can be grouped into three types: Dictyostelid, Polysphondylid, and Acytostelid forms [4].

Historically, the classification of dictyostelids into genera was based on three sorocarpic features, leading to the recognition of three genera: *Dictyostelium*, *Polysphondylium*, and *Acytostelium* [4]. In addition to sorocarpic structures, other features such as spore characteristics and patterns of aggregation and streaming were also used to distinguish genera [5]. Later, the use of 18S rDNA sequences altered dictyostelid classification. Phylogenetic analyses based on this genetic marker revealed 4 [6], 8 [7], and eventually 12 [5] distinct monophyletic clades, underscoring the need for taxonomic reclassification of these highly diverse organisms.

Twelve genera: *Dictyostelium*, *Polysphondylium*, *Coremiostelium*, *Raperostelium*, *Hagiwaraea*, *Tieghemostelium*, *Speleostelium*, *Synstelium*, *Heterostelium*, *Rostrostelium*, *Acytostelium*, and *Cavenderia*, were proposed based on combined morphological features and 18S rDNA sequence data [5]. These genera were grouped into four families: Dictyosteliaceae, Raperosteliaceae, Acytosteliaceae, and Cavenderiaceae. The first two families belong to the order Dictyosteliales, while the latter are classified under the order Acytosteliales. Notably, no family was assigned to *Coremiostelium* and *Synstelium*. Additionally, Sheikh et al. [5] identified unique signature sequences within the 18S rDNA gene that can be effectively used for genus-level classification, further enhancing the resolution of dictyostelid taxonomy.

Recent studies on the diversity of dictyostelids have employed a combination of molecular, morphological, and ecological characteristics for species identification [8,9,10,11,12]. While the 18S rDNA sequence has primarily been used for genus-level identification, morphological features such as the structure of fruiting bodies, including the shape, size, and color of stalks and spores, have played a key role in distinguishing species [8,9,10,12]. Additional traits, including the size of sorous, aggregation pattern, and the form of pseudoplasmodia, have also been critical for species-level classification [8,12]. Furthermore, ecological factors, such as soil type and environmental conditions, have been considered in identifying new species [8,12]. These findings highlight the necessity of integrating data from multiple approaches for accurate species identification in dictyostelids.

The 18S rDNA, which encodes the small subunit (SSU) rRNA gene, has been used for classifying dictyostelids at the genus level [5]. However, its effectiveness for species-level classification has not been investigated. Rukseree et al. [11] examined the use of partial 18S rDNA sequences to reveal the diversity of dictyostelids found in Amnat Charoen Province, Thailand. Their results identified five genera: *Cavenderia*, *Heterostelium*, *Raperostelium*, *Dictyostelium*, and *Polysphondylium*. However, no species-level classification was reported.

Besides the 18S rDNA, sequences of other regions and DNA marker techniques were applied to study the diversity and evolution of dictyostelids, indirectly suggesting DNA markers for species-level classification. The 5.8S ribosomal DNA and internal transcribed spacer (ITS) were utilized to classify 28 samples across seven species. Phylogenetic trees indicated that all species, except for *D. mucoroides*, formed monophyletic clades that corresponded with defined species [13]. Additionally, protein sequences from six genes—*agl*, *amdA*, *purD*, *purL*, *rpaA*, and *smdA*—were analyzed for phylogenetic purposes. The resulting phylogeny showed topologies aligned with new genus classifications, although the clades for *Raperostelium* species were non-monophyletic [14]. For the DNA marker technique, single-strand conformational polymorphism fingerprinting (SSCP) was developed to study the genetic diversity of 73 Thai dictyostelid isolates. The SSCP fingerprint patterns were consistent with genus-level taxonomy derived from SSU rDNA phylogenies [15]. Besides the study on 5.8S rDNA and ITS sequence limited to seven species [13], no studies have examined intraspecific variation, leaving unanswered questions regarding which genomic region or marker technique might be effective for dictyostelids species-level classification.

Classifying dictyostelid species based on morphological features is time-consuming and requires specialized expertise. Using DNA markers for species-level classification offers an alternative approach, which would be helpful, particularly in amplicon-based metagenomic studies, where rapid and accurate identification is crucial for understanding microbial diversity. Although the small subunit ribosomal RNA gene (18S rDNA) has been widely used for genus-level identification of dictyostelids [5], its sole application for species-level classification and the associated limitations remain underexplored. Therefore, this study aims to evaluate the effectiveness and limitations of using 18S rDNA sequences to distinguish dictyostelids at the species level. We hypothesize that while the marker may show utility in specific genera, it will also reveal the underlying reasons for its limitations in achieving accurate species-level classification across dictyostelids genera. 

## 2. Materials and Methods

### 2.1. Sample Collection and 18S rDNA Amplification

Fourteen dictyostelid samples found in Thailand, *D. discoidium* and *D. purpureum*, were obtained from the Dicty Stock Center, Northwestern University (http://dictybase.org/StockCenter/StockCenter.html accessed on 11 February 2022). The 14 samples, representing 14 species, were naturally isolated, with collection and isolation details for eight samples described in related published literature [8,12]. The geographical distribution of these samples is detailed in Table 1, which includes specific collection sites in Thailand. *D. discoideum* and *D. purpureum* were included as positive controls for 18S rDNA amplification and analysis. They were cultured using *Escherichia coli* (DH5-Alpha) on SM agar under 23–25 °C for 4–7 days until they developed sorocarps (Figure 1). The partial 18S rRNA gene was directly amplified from fruiting body using 18S_FA (AAC CTG GTT GAT CCT GCC AG) and 18S_RB (TGA TCC TTC TGC AGG TTC AC) primers [11,16] with Q5 High-Fidelity DNA polymerase (New England Biolabs, Ipswich, MA, USA). PCR process included pre-denaturation at 98 °C for 2 min., then performing 30–35 rounds of denaturation at 98 °C for 30 s, annealing at 60 °C for 40 s, and extension at 72 °C for 1.10 min. The final extension was performed at 72 °C for 2 min. The PCR products were dissolved in agarose gel electrophoresis, and the band was excised and gel-purified before submitting to a sequencing company (U2Bio, Seoul, Republic of Korea) for Barcode Taq sequencing (BTSeq^TM^).

### 2.2. Sequence Data Collection 

A list of dictyostelid genera was first retrieved from the NCBI taxonomy database using the keyword “dictyostelia”. This list was supplemented with additional keywords such as “18S”, “18S rRNA”, and “18S ribosomal RNA” to search for relevant 18S rDNA sequences in the NCBI nucleotide database (GenBank). Based on published literature, we verified and updated species names for sequences labeled as unidentified or undescribed. Sequences that lacked species identification were excluded from the dataset.

The retrieved sequences were aligned using MAFFT (version 7) [17] with default parameters to ensure consistency across the dataset. As part of the quality control process, sequences that were unusually short or markedly divergent from others within the same species or genus were excluded. The resulting dataset comprised 309 sequences representing 150 species across 12 genera, with the accession numbers documented and compiled using R programming (version 4.2.1) [18], as detailed in Appendix A. The 18S rDNA sequence of *Acanthamoeba castellanii* (U07401) was collected as an outgroup for phylogenetic analysis.

### 2.3. 18S rDNA Sequence Analysis 

The sequences of PCR products amplified from 16 dictyostelids samples were compared to the GenBank nr database to confirm their authenticity as 18S rDNA sequences of dictyostelids, using blastn (https://blast.ncbi.nlm.nih.gov/Blast.cgi accessed on 18 January 2024) [19]. The genus of each sample was classified according to a signature sequence, following the method proposed by Sheikh et al. [5]. These sequences were compared to those in the collected database sequences using MAFFT (version 7) [17] and manually edited in AliView (version 1.28) [20]. The aligned sequences were used for both distance-based and phylogeny-based species delimitation analyses. This study’s distance-based species delimitation methods included comparisons of intraspecific and interspecific distances, barcoding gap analysis, and assembling species by Automatic Partitioning (ASAP) [21]. The maximum likelihood phylogenetic analysis and Bayesian implementation of the Poisson tree process model (bPTP) [22] were applied as phylogeny-based species delimitation approaches.

For distance-based analyses, the aligned 18S rDNA sequences were separated into two datasets: one for calculating intraspecific distances and the other for interspecific distances, ensuring no overlap between the two. The interspecific distances were further categorized into “within-genus” and “between-genus” groups. Distances were calculated using Kimura’s 2-parameter (K2P) model. Kruskal–Wallis rank sum and pairwise Wilcoxon tests were performed to assess whether the average distances between these groups showed significant differences. These statistical analyses and visualizations were conducted using R software (version 4.2.1) [18].

Additionally, the aligned sequences used for intraspecific distance calculations were applied to barcoding gap and ASAP (Assemble Species by Automatic Partitioning) analyses, performed using the online ASAP tool (https://bioinfo.mnhn.fr/abi/public/asap/asapweb.html accessed on 25 October 2022) [21]. Barcoding gaps were determined as the difference between the minimum interspecific K2P distance and the maximum intraspecific K2P distance, calculated using the nonConDist and maxInDist functions of the R package *Spider* (version 1.5.0) [23]. When the barcoding gap value is less than or equal to zero, the minimum interspecific K2P distance is equal to or smaller than the maximum intraspecific K2P distance, making it impossible to define a threshold to separate within-species and between-species variation. In contrast, when the barcoding gap value is positive, the minimum interspecific K2P distance exceeds the maximum intraspecific K2P distance, enabling species boundaries to be distinguished based on sequence variation.

The Wilcoxon rank sum test was used to evaluate whether the average barcoding gap values were significantly different from zero. Due to the limited number of samples per species, barcoding gap values were grouped by genus. The Kruskal–Wallis rank sum test was also employed to determine whether average barcoding gaps significantly differed among genera. The statistical analyses and visualization were performed using R (version 4.2.1) [18].

For the ASAP analysis, distances were computed using the Kimura 2-parameter model (K80) with a transition/transversion (ts/tv) ratio of 2.0, while other parameters were set to default values.

For phylogeny-based analyses, the maximum likelihood (ML) phylogeny was constructed using the raxmlGUI (version 2.0.9) [24] based on the aligned 18S rDNA sequences used to calculate intraspecific distances. The optimal evolutionary model for the ML phylogeny was selected using the Bayesian Information Criterion (BIC), calculated through the *run modeltest* function in raxmlGUI (version 2.0.9). The model with the lowest BIC value was chosen as the most appropriate for the data. The phylogenetic tree was rooted using *Acanthamoeba castellanii* as an outgroup. One thousand bootstrap replicates were performed to ensure statistical robustness, providing confidence levels for each clade.

The resulting phylogenetic tree was then used as input for bPTP analysis via the online platform (https://species.h-its.org accessed on 25 October 2022) [22]. The bPTP parameters included 500,000 Markov Chain Monte Carlo (MCMC) generations, a thinning interval of 500, and a burn-in of 10% [25]. The phylogenetic tree was visualized and manually edited for presentation using FigTree software (version 1.4.4) [26].

### 2.4. Success Rate of Species Delimitation Calculation

The success rate of species delimitation using the 18S rDNA sequence was evaluated based on the outcomes from the ASAP, ML phylogeny, and bPTP methods. For the ML phylogeny, species relationships were assessed based on monophyly for species with multiple samples. Species were considered successfully delimited if all samples belonging to a species formed a monophyletic clade. The success rate was calculated as the proportion of monophyletic species among all species with multiple samples.

For the ASAP and bPTP methods, relationships among samples of the same species were categorized as MONO, MERGE, or SPLIT. MONO indicated that all samples of a species clustered into a single group. MERGE referred to cases where samples of a species were grouped with samples from other species. SPLIT was used when samples of a species were divided into multiple groups. Additionally, the term multi-SPLIT was assigned to species whose samples were distributed across multiple groups, with each group carrying members of at least two species.

The success rate for the ASAP, ML phylogeny, and bPTP methods was calculated as the proportion of species assigned as MONO among all species including ones with multiple samples. A weighted success rate, which accounted for the number of samples per species, was also computed to provide a refined assessment.

## 3. Results

The nearly complete 18S rRNA gene from 16 selected samples was amplified and sequenced using the BTSeq™ method to assess the limitations of using 18S rDNA sequences for species-level classification in dictyostelids. The identity of these sequences as dictyostelid 18S rDNA was confirmed through blastn analysis and signature sequence verification. These sequences were combined with 310 database entries and analyzed using barcoding gap analysis and multiple species delimitation approaches, including ASAP, maximum likelihood (ML) phylogeny, and bPTP.

### 3.1. Identification of 16 Selected Samples

The 18S rDNA sequences obtained in this study were submitted to the GenBank database under accession PQ834586-601. The quality of the 18S rDNA sequences obtained in this study was assessed based on the depth of coverage and blastn analysis. The depth of coverage ranged from 1 to 210, and blastn results confirmed that the sequences belonged to dictyostelids, precisely 15 *Dictyostelium* samples and 1 *Raperostelium* sample (Table 1). Interestingly, some blastn results differed from the genus classifications previously defined in the Dicty Stock Center (http://dictybase.org/StockCenter/StockCenter.html accessed on 18 April 2024). For example, TH14B (Delta) and TH18B sequences were classified as *Dictyostelium* sp. and *Polysphondylium* sp., respectively, in the Dicty Stock Center. However, blastn analysis revealed that these sequences were similar to *Raperostelium monochasioides* (AM168052.1) and *Dictyostelium* sp. *THC11X* (HQ141523.1), respectively (Table 1).

Additionally, conflicts were observed for four samples previously classified as *Cavenderia* (*C. subdiscoidea*, *C. pseudoaureostipes*, *C. bhumiboliana*, and *C. protodigitata*) [12]. Genus-specific signature sequence analysis [5] was conducted to address these inconsistencies. For four samples of *Dictyostelium*, this analysis was consistent with both the Dicty Stock Center classification and blastn results (Table 1 and Figure 2). The eight samples previously classified as *Cavenderia*, either in the Dicty Stock Center or published studies [8,12], also showed agreement between their original classification and the results of signature sequence analysis (Table 1 and Figure 2). However, three samples (TH8C, TH11CW, and TH14B) exhibited disagreements between blastn results and signature sequence analysis. While blastn suggested different genera, signature sequence analysis classified these samples as belonging to *Raperostelium* (Table 1 and Figure 2).

These findings highlight the use of signature sequence analysis as a reliable method for genus classification [5], especially when discrepancies arise between different classification methods.

### 3.2. Interspecific and Intraspecific Variation Among Selected Samples

Due to the similarity in blastn results for some samples, particularly among *Cavenderia* and *Dictyostelium* species, where sequences showed comparable levels of identity to only a limited number of database entries, the 18S rDNA sequences from 16 samples were analyzed for pairwise similarity. As summarized in Table 2, ten haplotypes were identified among the 16 samples. Notably, TH11C and TH18CC, TH14B and TH11CW, as well as a group comprising *C. ungulata*, *C. helicoidea*, *C. protumula*, *C. bhumiboliana*, and *C. protodigitata*, were represented by three distinct haplotypes. The proportion of pairwise differences and Kimura 2-parameter (K2P) distances between species ranged from 0.000 to 0.265 and 0.000 to 0.330, respectively. The average interspecific K2P distance within a genus (0.012) was significantly lower compared to the interspecific distances between genera (0.281) and between orders (0.321).

The eight *Cavenderia* samples were previously classified, and their corresponding 18S rDNA sequences were submitted to the database under accessions, as shown in Appendix A [8,12]; however, only three samples: *C. subdiscoidea*, *C. pseudoaureostipes*, and *C. bhumiboliana*, showed blastn best hit corresponded with their published accession numbers (Table 1). In order to explore why the blastn best hits of our five samples were not their corresponding published data, the published 18S rDNA sequences for eight *Cavenderia* species were compared to those obtained in this study (Appendix A). The results revealed minimal variation within species, with degrees of difference ranging from 0.000 to 0.001. Among the group of our samples comprising *C. ungulata*, *C. helicoidea*, *C. protumula*, *C. bhumiboliana*, and *C. protodigitata* presenting one haplotype (Table 2), pairwise distances between species were consistently 0.000, except for the comparisons involving *C. helicoidea*, which displayed distances greater than zero. The average interspecific distances calculated either from the published database or the sequences obtained in this study were consistent at 0.003.

These findings indicate overlapping ranges of interspecific and intraspecific differences and K2P distances, highlighting the limitations of using the 18S rDNA sequence for species-level classification. 

### 3.3. Interspecific and Intraspecific Variation Comparison and Barcoding Gap Analysis

The 309 rDNA sequences from 149 species retrieved from the GenBank database were combined with 16 sequences from our samples, aligned, and applied to calculate pairwise differences and Kimura 2-parameter (K2P) distances calculation, which were then categorized into between-genus interspecific, within-genus interspecific and intraspecific variations (Figure 3A). A comparison of the average statistics among these three groups revealed significant differences (Kruskal–Wallis rank sum test, *p* < 0.001). Pairwise comparisons using the Wilcoxon rank sum test showed significant differences between intraspecific and interspecific variations and between within-genus and between-genus interspecific variations (*p* < 0.001). However, the K2P distance ranges between these groups were not clearly separated. Notice that the K2P distance range for between-genus interspecific variation was higher than that of within-genus interspecific variation, and the minimum interspecific variation and intraspecific variation were identical.

Because the data used for calculating interspecific variation comprised only species carrying a single sample, only the intraspecific K2P distances were further applied for barcoding gap calculation. The barcoding gaps were then divided into nine groups based on genus. As shown in Figure 3B, five genera—*Raperostelium*, *Coremeostelium*, *Synstelium*, *Rostrostelium*, and *Acytostelium*—displayed positive barcoding gaps. This indicates that, for these genera, the minimum interspecific distance exceeded the maximum intraspecific distance, allowing for the potential determination of a threshold to distinguish variation within and between species. The Wilcoxon rank sum test indicated that the average barcoding gaps for four genera—*Polysphondylium*, *Raperostelium*, *Heterostelium*, and *Cavenderia*—significantly differed from zero. Only *Raperostelium* exhibited positive and statistically significant average barcoding gap values among these. Furthermore, the Kruskal–Wallis rank sum test also demonstrated significant differences in average barcoding gaps across genera (*p* < 0.001).

These findings support the utility of 18S rDNA for genus-level classification but reveal its limitations for reliable species-level classification due to overlapping distance ranges and the wide variation in the number of samples among genera.

### 3.4. Species Delimitation Analysis Using ASAP

The alignment previously used for barcoding gap analysis was further applied to similarity-based species delimitation using the ASAP method. The analysis did not include two *Dictyostelium* sp. and four *Raperostelium* sp. because their species names were not assigned. ASAP classified the 214 samples of 44 species into 48 groups, of which only 10 corresponded to one species. Species with at least 10 samples, such as *Cavenderia aureostipes*, *C.* cf. *aureostipes*, *Dictyostelium discoideum*, *D. purpureum*, and *Heterostelium pallidum*, were split into multiple groups, displaying SPLIT or multi-SPLIT patterns (Table 3, Figure 4, Figure 5 and Figure 6). In contrast, species with fewer than 10 samples were often merged with others. This pattern of merged species within the same genus agreed with the observed overlap between intraspecific and within-genus interspecific variation (Figure 3A).

Notably, no merged species were observed in genera with high barcoding gaps, such as *Raperostelium*, *Coremiostelium*, *Synstelium*, *Rostrostelium*, and *Acytostelium* (Figure 3B). Among our samples, *Dictyostelium discoideum* and *D. purpureum* were grouped with most of their corresponding species samples group 1 and 12. Samples belonging to *Cavenderia* were divided into two groups by ASAP, with all except *C. subdiscoidea* clustered together in group 45. This classification was consistent with the blastn results (Table 1), as most *Cavenderia* samples, except *C. subdiscoidea* and *C. pseudoaureostipes*, matched a single recorded sequence. These findings also partially aligned with the genetic distances among *Cavenderia* samples (Table 2), where zero pairwise distances indicated a shared haplotype among five species (*C. ungulata*, *C. helicoidea*, *C. protumula*, *C. bhumiboliana*, and *C. protodigitata*).

The success rate for species delimitation using ASAP was 22.73% overall and 12.15% when weighted by species count. These results suggest that the interspecific distances were often insufficient to separate taxa using the ASAP method reliably.

### 3.5. Species Delimitation Analysis Using ML Phylogeny

The alignment used for ASAP analysis was also subjected to maximum likelihood (ML) phylogenetic analysis, which revealed nine monophyletic clades corresponding well to genus-level classifications (Figure 4A). Of the 44 previously defined species, 19 formed monophyletic clades with strong bootstrap support (Figure 4, Figure 5 and Figure 6, Table 3). These 19 species had fewer than 10 samples each, except for *Dictyostelium purpureum*. In contrast, species with more than 10 samples, such as *D. discoideum*, exhibited polyphyletic relationships. Interestingly, all *D. discoideum* samples, except one (AM168039), formed a monophyletic group, consistent with the ASAP species delimitation analysis results.

For species with fewer than 10 samples, 21 displayed non-monophyletic relationships (Table 3), often clustering with samples from other species to form monophyletic groups (Figure 4, Figure 5 and Figure 6). This pattern of non-monophyletic relationships aligned with the overlap between intraspecific and within-genus interspecific variation observed in Figure 3A. Monophyletic relationships were observed in all species from genera with high barcoding gaps (Figure 3B), except *Raperostelium tenue*.

Among our samples, *D. discoideum* and *D. purpureum* clustered with their respective species and formed monophyletic clades. For the eight species of *Cavenderia*, only two formed monophyletic clades. However, these eight *Cavenderia* species and some samples of *C. aureostipes* and *C.* cf. *aureostipes* were clustered into a larger monophyletic clade with 80% bootstrap support. Six *Cavenderia* samples (except for *C. subdiscoidea* and *C. pseudoaureostipes*) formed a separate monophyletic clade with 98% bootstrap support within this clade. This grouping corresponded well with the single blastn best-hit results for these samples (Table 1). It also aligned with the genetic distance data (Table 2), which showed these five species sharing a single haplotype.

The success rate for species delimitation using ML phylogenetic analysis was 43.18%, dropping to 31.31% when weighted by sample number. These findings suggest that ML phylogenetic analysis is more effective for species-level classification than similarity-based approaches.

### 3.6. Species Delimitation Analysis Using bPTP

The maximum likelihood (ML) phylogeny was used as the basis for a bPTP (Bayesian Poisson Tree Processes) analysis, a phylogeny-based approach for species delimitation. The bPTP results assigned the 214 sequences from 44 species into 27 putative taxa (Figure 4, Figure 5 and Figure 6, Table 3). Among these 27 taxa, only three—*Raperostelium ellipticum*, *Heterostelium asymetricum*, and *H. oculare*—consisted exclusively of single-species samples. For species with more than 10 samples, all except *H. pallidum* were merged with samples from other species within the same genus. For example, *Dictyostelium discoideum* and *D. purpureum* were grouped with other *Dictyostelium* species into a single taxon (group no. 7). In contrast, *H. pallidum* exhibited a multi-SPLIT pattern, with the samples divided into two groups and merged with samples from other species.

Most species with fewer than 10 samples were merged into a small number of putative taxa (Table 3, Figure 4, Figure 5 and Figure 6), except *H. candidum*, which presented the same relationship pattern among samples as *H. pallidum*. This extensive merging of species within the same genus corresponded with the overlap between intraspecific and within-genus interspecific variation observed in Figure 3A. However, no merged species were found in genera with high barcoding gaps, such as *Raperostelium*, *Coremiostelium*, *Synstelium*, *Rostrostelium*, and *Acytostelium*. In contrast, merged species were prevalent in genera with low barcoding gaps (Figure 3B), such as *Dictyostelium* (Figure 4) and *Cavenderia* (Figure 6), where all samples within each genus were assigned to a single putative taxon. The classification of our *Cavenderia* samples into a single putative taxon was consistent with the blastn results (Table 1) and the genetic distance data (Table 2), as most *Cavenderia* samples shared the same haplotype.

The success rate of species delimitation using the bPTP method was 6.82% overall, decreasing to 2.80% when weighted by the number of species. These findings indicate that the bPTP method had the lowest resolution for species delimitation among the methods used in this study and was the least effective.

## 4. Discussion

Dictyostelid species identification based on morphology requires experienced taxonomists due to the subtle differences in their traits [5,8,10,27,28]. DNA markers offer a promising alternative to accelerate species identification and facilitate studies of their diversity, particularly in environmental samples. However, no universally effective genetic marker has yet been established for these organisms. The small subunit ribosomal RNA gene (18S rDNA) has been widely used for classifying social amoebae [6,7] and finally leading the new classification of dictyostelids at the genus level [5]. However, its potential for species-level classification remains uncertain. This study aimed to evaluate the capabilities and limitations of the 18S rDNA sequence for species-level classification in dictyostelids. Our findings confirmed its effectiveness for genus-level classification and demonstrated its potential for species-level classification in specific genera. Furthermore, comparing success rates across different approaches highlighted maximum likelihood (ML) phylogenetic analysis as the most effective method for species-level classification in this context.

A comparison of the blastn and signature sequence analyses for our samples revealed that all genus-level classifications were correctly assigned using the signature sequence approach (Table 1). This high accuracy may stem from many new genera, such as *Raperostelium* and *Cavenderia*, initially proposed based on 18S rDNA sequences [5]. Additionally, the topology of the maximum likelihood (ML) phylogeny, constructed using these sequences alongside database sequences, showed monophyletic clades that aligned well with genus classifications (Figure 4A), further validating the signature sequence analysis. Similarly, a previous 18S rDNA phylogeny built from various species across 12 dictyostelid genera [27] demonstrated a topology consistent with the genus classifications proposed by Sheikh et al., 2018 [5]. This sequence-based genus classification was also supported by the observed differences between between-genus and within-genus interspecific variation (Figure 3A) and by the results of species delimitation analyses (Figure 4, Figure 5 and Figure 6). Together, these findings reinforce the utility of the 18S rDNA sequence as a reliable tool for genus-level classification in dictyostelids.

Although the 18S rDNA sequence proves effective for genus-level classification and identification, several factors highlight its limitations for species-level classification. These include the low number of haplotypes observed in our samples (Table 1 and Table 2), the overlap between interspecific and intraspecific variations (Figure 3A), the negative barcoding gap in specific genera (Figure 3B), and the merging of multiple species in the species delimitation analyses (Figure 4, Figure 5 and Figure 6, Table 3). These findings suggest that the high diversity of 18S rDNA sequences within species reduces its utility for precise species identification.

Previous studies on dictyostelid 18S rDNA sequences have primarily used this marker to classify social amoebae into broader groups [5,6,14,29], but few have compared genetic variations within and between species. Recent studies examining dictyostelid diversity in China incorporated multiple samples of species like *D. discoideum*, *D. purpureum*, and *H. pallidum* into phylogenetic analyses, showing that only a subset of single-species samples formed monophyletic clades. This further highlights the limitations of 18S rDNA sequences for species-level resolution [27,28]. Another study focused on soil eukaryote diversity using a portion of the 18S rDNA gene successfully distinguished *Amoebozoa* from other protists but did not provide detailed resolution at lower taxonomic levels. The authors suggested that this may reflect the relatively limited molecular studies on protists compared to animals, plants, and fungi [30].

An attempt to use 18S rDNA sequences to explore the diversity of eukaryotes, especially from environmental samples, led to the detailed characterization of this sequence. The 18S rDNA gene features nine variable regions (V1–V9), of which V2, V4, and V9 are deemed most suitable for biodiversity assessments, as they define the highest number of genera (~50% at 97% sequence similarity) [31]. This sequence has provided a broad framework for classifying eukaryotes at higher taxonomic levels, including fungi and amoebozoans. For fungi, adding ITS regions as genetic markers offers significantly finer resolution, with ITS sequences at 97–99% similarity, reliably supporting species-level classification [32,33]. In contrast, similar criteria for species-level classification have not been clearly established for dictyostelids. Consequently, no studies have conclusively demonstrated the effectiveness of 18S rDNA sequences for species delimitation in dictyostelids.

Specific genera showed positive barcoding gaps (Figure 3B), presented monophyletic relationships among samples (Figure 5), or were assigned to putative taxa corresponding to their species (Table 3), suggesting the possible use of this sequence for species-level classification of these genera. These genera included *Raperostelium*, *Coremiostelium*, *Synstelium*, *Rostrostelium*, and *Acytostelium,* which carried one or more characteristics mentioned above. *R. ellipticum* exhibited a large barcoding gap and consistently formed a distinct group (Table 3).

These results indicate the potential of 18S rDNA sequences for species-level classification in specific dictyostelid genera. However, potential sources of error and imbalanced sample sizes could significantly influence the effectiveness of barcoding gaps and species delimitation analyses. Two possible sources of error related to the results of this study were the reliability of morphology-based species classification and post-alignment sequence processing.

While dictyostelid species are typically classified using molecular phylogenetic data and morphological characteristics, morphological traits are often the primary basis for defining new taxa, with molecular data serving as supplementary evidence [8,12]. This approach can introduce uncertainty, particularly when intraspecific morphological variation is documented but corresponding molecular variation is not. For instance, samples of *Cavenderia* cf. *aureostipes* were morphologically identified as *C. aureostipes*, but phylogenetic analyses suggested that this species complex requires taxonomic revision [34]. Another potential source of error is related to post-alignment sequence processing, specifically the trimming of aligned sites to remove indels possibly arising from missing data. While this step helps maintain data quality, it may also reduce sequence diversity. However, as only a small proportion of data is typically lost during trimming, its impact on the barcoding gap and species delimitation analyses is likely minimal.

Sampling bias, regarding the number of species per genus and samples per species, presents another significant challenge. As shown in Figure 4, Figure 5 and Figure 6 and Table 3, some genera, such as *Coremiostelium*, *Synstelium*, and *Rostrostelium*, contain fewer than five species, whereas genera like *Dictyostelium*, *Heterostelium*, and *Cavenderia* include more than 10 species. Similarly, the number of samples per species varies widely; for example, only five species from three genera have at least 10 samples per species (Table 3). These sampling biases generated a combination of data from species with sufficient samples to capture intraspecific variation and those with too few samples, which can obscure the distinction between interspecific and intraspecific variation by barcoding gaps analysis. These biases can also lead to over-splitting or lumping of species boundaries by species delimitation analysis [35,36].

To evaluate the effectiveness of selected species delimitation methods, we compared their success rates, with ML phylogenetic analysis yielding the highest rate and bPTP showing the lowest (Figure 4, Figure 5 and Figure 6 and Table 3). However, none of the methods achieved a success rate exceeding 50%, supporting the conclusion that the 18S rDNA sequences have limited capability for species-level classification. All three approaches accurately classified samples of three species—*Rostrostelium ellipticum*, *Heterostelium asymetricum*, and *Heterostelium oculare*—into the correct number of species. Additionally, the other seven species represented by 2–4 samples were correctly assigned into distinct taxa by both ASAP and ML phylogenetic analysis. Notably, species with at least ten samples were divided into multiple taxa by ASAP or formed non-monophyletic clades in the ML phylogeny. An exception was *D. purpureum*, which formed a monophyletic clade. Regardless of the number of samples per species, most species were grouped with other samples from the same genus as a single taxon in the bPTP analysis. Interestingly, branch length appeared to influence species-level classification in the bPTP method; for example, all *Dictyostelium* and *Cavenderia* samples were grouped into just two taxa each, reflecting the relatively short branch lengths within the monophyletic clades of these genera. These findings suggested that the number of samples per species and method selection influence the number of assigned taxa. In addition, the ML phylogenetic analysis is this study’s most reliable method for species-level classification.

As noted earlier, none of the species delimitation methods used in this study achieved a success rate above 50%. The low success rates are likely influenced by sampling bias arising from limited species diversity and a small number of samples per species [35]. For ASAP, overlapping interspecific and intraspecific distances would reduce its effectiveness, as it relies on a clear threshold for distinguishing species. The overlapping distances result in uncertain boundaries, leading to errors such as over-splitting (where a single species is mistakenly divided into multiple units) or lumping (where multiple species are incorrectly grouped as one). Future studies should minimize sampling bias by including a broader range of species and increasing the sample size for species currently underrepresented in the dataset to increase the success rate of species delimitation analysis.

## 5. Conclusions

DNA markers hold great potential for reducing the cost and time required to study the diversity of dictyostelids, particularly in environmental samples. However, no highly effective DNA marker has yet been proposed. While the 18S rDNA sequence has been widely used to investigate dictyostelids’ diversity and evolutionary relationships, prior studies have primarily focused on diversity at the species level, often using only one sample per species. This study aimed to evaluate the strengths and limitations of the 18S rDNA sequence for species-level classification. The results supported its use for genus classification and highlighted its potential, based on barcoding gap and species delimitation analyses, for distinguishing species within specific genera, such as *Raperostelium* and *Rostrostelium*. Among the methods tested, ML phylogenetic analysis emerged as the most effective for species-level classification.

Nevertheless, the analyses were constrained by the limited number of samples per species and the uneven representation of species across genera, which reduced the statistical power. Future studies should include more extensive and diverse sample sets spanning additional dictyostelid species to improve these methods’ reliability and identify more robust DNA markers for species-level classification.

## Figures and Tables

**Figure 1 microorganisms-13-00275-f001:**
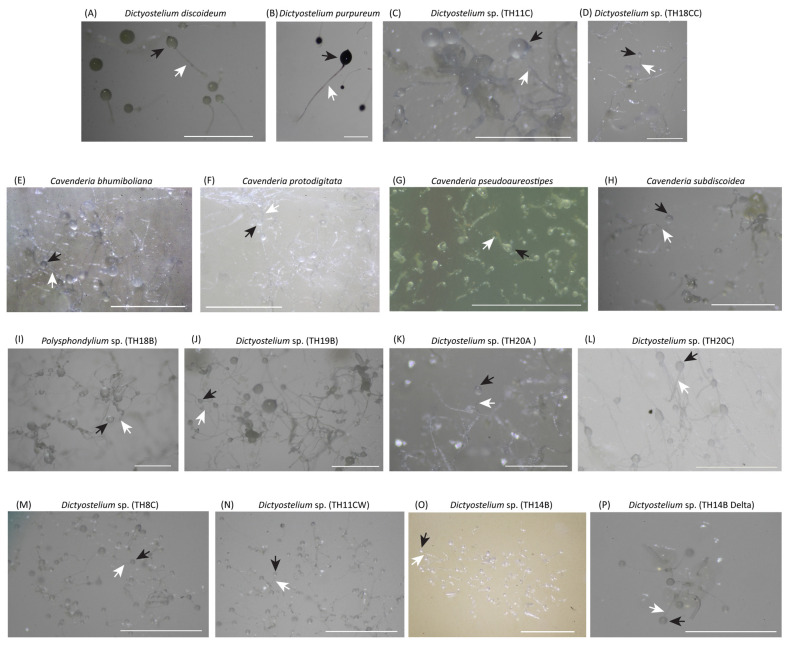
Sorocarps of 16 cultured samples comprised 11 *Dictyostelium* (**A**–**D**,**J**–**P**), 4 *Cavenderia* (**E**–**H**) and 1 *Polysphondylium* (**I**). Besides *D. discoideum* and *D. purpureum*, the other 14 samples were collected from Thailand and keep at the Dicty stock center. The black and white arrows indicated the sorus and the stalk. The scale is 1000 μm.

**Figure 2 microorganisms-13-00275-f002:**
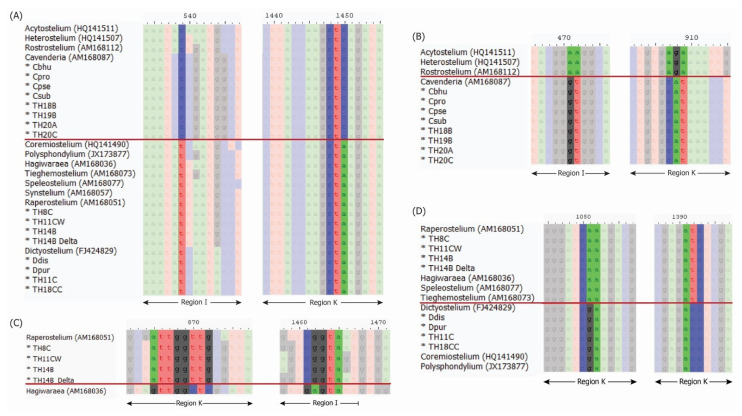
Signature sequence of 18S rRNA gene used for genus classification by Sheikh et al. 2018 [5]: comparison between Acytosteliales and Dictyosteliales (**A**), comparison between Acytosteliaceae and Cavenderiaceae (**B**), comparison between *Raperostelium* and *Hagiwaraea* (**C**), and comparison between Raperosteliaceae and Dictyosteliaceae (**D**). The red line in each subfigure separates the two groups of compared samples. The positions on these signature sequences are based on combining selected sample sequences to the alignments reported by Sheikh et al. 2018 [5].

**Figure 3 microorganisms-13-00275-f003:**
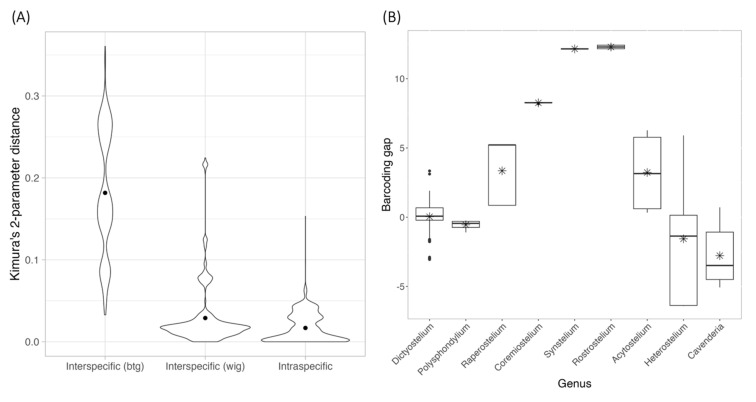
The comparison between interspecific and intraspecific K2P distance of 18S rDNA sequences (**A**) and the barcoding gap distribution (**B**). The interspecific distance values were divided into two groups: within-genus (wtg) and between-genus (btg) interspecific distance. The dot inside each violin plot and the star inside each box-and-whisker plot presented the mean of the K2P distance and barcoding gap.

**Figure 4 microorganisms-13-00275-f004:**
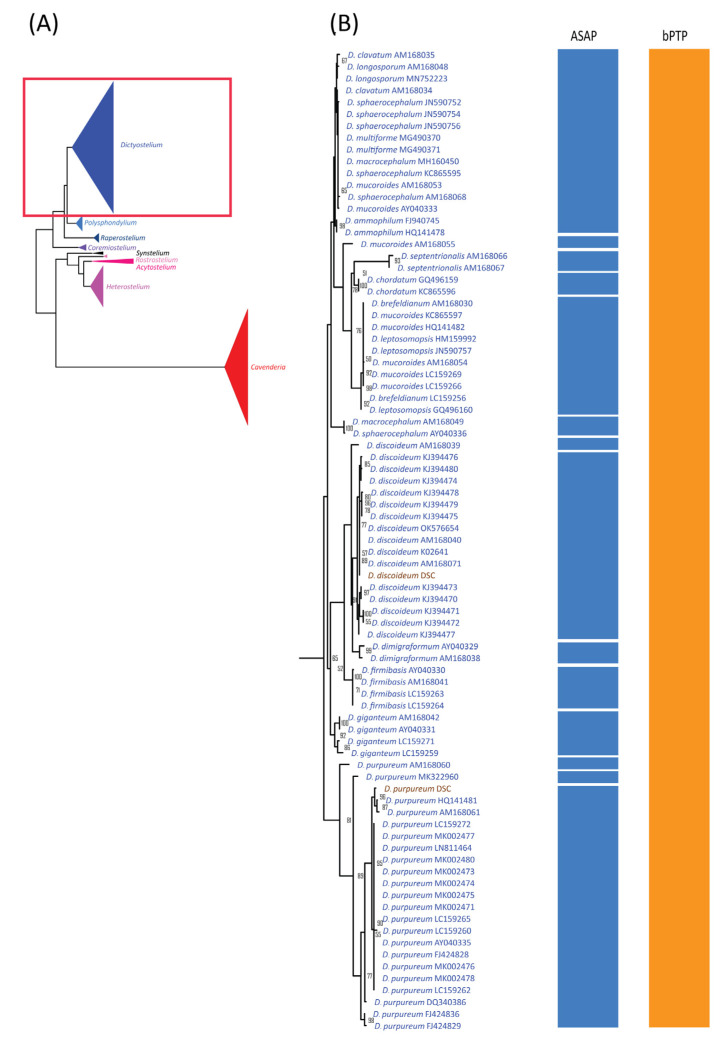
Comparison of the maximum likelihood (ML) phylogeny of *Dictyostelium* 18S rDNA sequences with species delimitation results from ASAP and bPTP. The ML phylogeny was constructed using the GTR + I + G model with 1000 bootstrap replicates (**A**). The ML phylogeny and species delimitation results were presented in four sections, with this figure showing the first section: samples belonging to the genus *Dictyostelium* (**B**). Samples highlighted in brown text represent cultured samples. Only bootstrap support values greater than or equal to 50% are shown.

**Figure 5 microorganisms-13-00275-f005:**
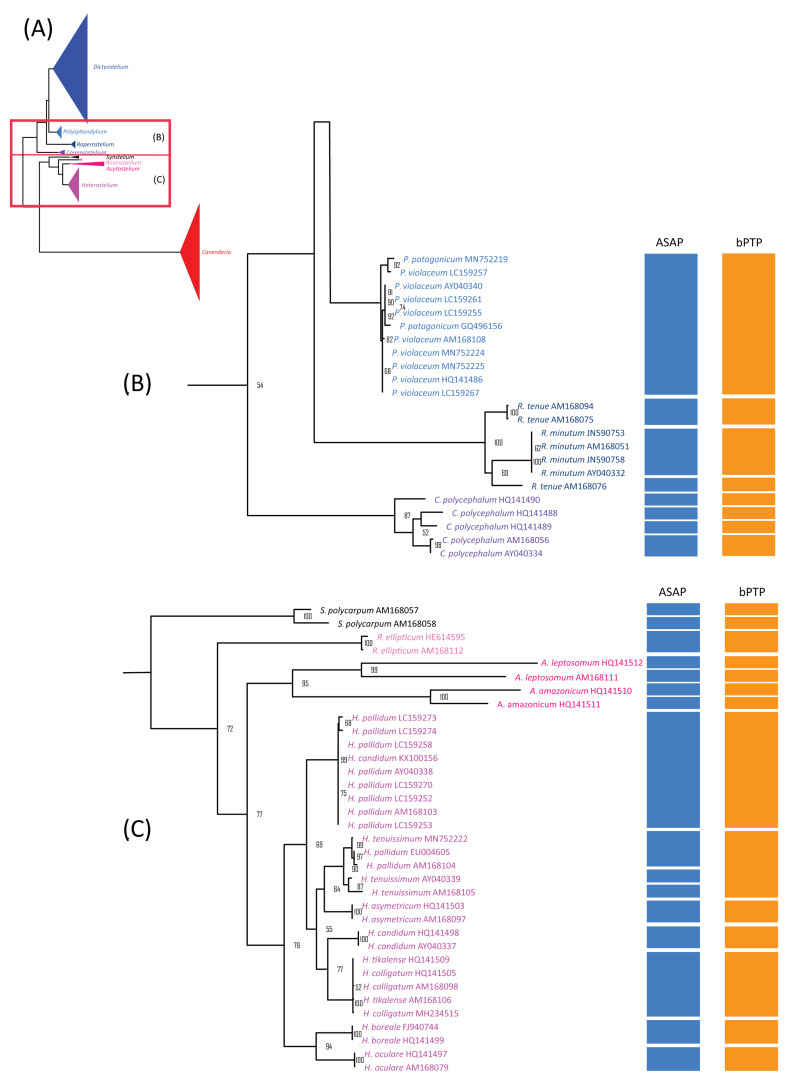
The maximum likelihood (ML) phylogeny of dictyostelid 18S rDNA sequences was compared to the species delimitation results from ASAP and bPTP. The ML phylogeny was constructed using the GTR + I + G model with 1000 bootstrap replicates (**A**). The phylogeny and species delimitation results were divided into four sections, with the second and third sections shown here. Species from different genera were color-coded. The second section included samples from the genera *Polysphondylium* (blue), *Raperostelium* (dark blue), and *Coremiostelium* (purple) (**B**). The third section includes samples from the genera *Synstelium* (black), *Rostrostelium* (pink), *Acytostelium* (magenta), and *Heterostelium* (lavender) (**C**). Only bootstrap support values greater than or equal to 50% are shown.

**Figure 6 microorganisms-13-00275-f006:**
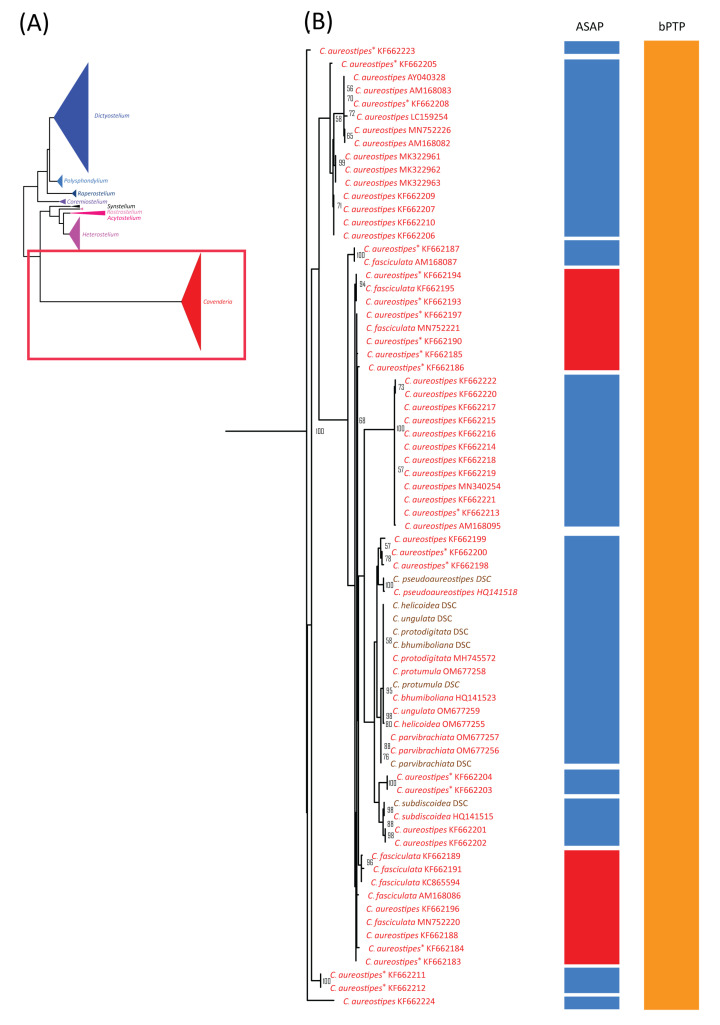
Comparison of the maximum likelihood (ML) phylogeny of dictyostelid 18S rDNA sequences with species delimitation results from ASAP and bPTP. The ML phylogeny was constructed using the GTR + I + G model with 1000 bootstrap replicates (**A**). The phylogeny and species delimitation results were divided into four sections, with the fourth section shown here. This section included samples from the genus *Cavenderia* (**B**). Samples in brown text indicate cultured samples. *C. aureostipes* with an asterisk (*C. aureostipes**) refers to *Cavenderia* cf. *aureostipes.* The red boxes highlight samples identified by ASAP as a single putative taxon. Only bootstrap support values greater than or equal to 50% are shown.

**Table 1 microorganisms-13-00275-t001:** Summary information of cultured samples: source identification, blastn result, sequence signature, and related publication. The e-value and query coverage of the blastn result was all 0.00 and at least 96%, respectively.

Sample Code	Previously Defined Species	Depth of Coverage	BLASTN Analysis			Genus Specified Based on Signature Sequence	Source as Stated in Dicty Stock Center	GenBank Accession No.
			Species	Accession Number	Percent Identity			
Dpur	*Dictyostelium purpureum* ^a^	2–115	*Dictyostelium purpureum*	AM168061.1	99.68%	*Dictyostelium*	Axenic strain grows in HL5 with 10% fetal bovine serum	PQ834598
Ddis	*Dictyostelium discoideum* ^a^	2–109	*Dictyostelium discoideum*	AM168071.1	100.00%	*Dictyostelium*	Wild strain obtained from Dennis Welker	PQ834601
TH11C	*Dictyostelium* sp. ^a^	1–192	*Dictyostelium mucoroides*, *Dictyostelium capitatum*	HQ141482.1, AM168032.1	99.84% 99.84%	*Dictyostelium*	Doi Inthanon National Park, Chom Thong District, Chiang Mai Province, Thailand	PQ834599
TH18CC	*Dictyostelium* sp. ^a^	3–184	*Dictyostelium mucoroides*, *Dictyostelium capitatum*	HQ141482.1, AM168032.1	99.84% 99.84%	*Dictyostelium*	Doi Inthanon National Park, Chom Thong District, Chiang Mai Province, Thailand	PQ834600
TH14B (Delta)	*Dictyostelium* sp. ^a^	1–210	*Raperostelium monochasioides*	AM168052.1	99.83%	*Raperostelium*	Songkhla Zoo, Mueang Songkhla District, Songkhla Province, Thailand	PQ834597
TH8C	*Dictyostelium* sp. ^a^	1–105	*Dictyostelium* sp. *TH8C*	HQ141492.1	99.89%	*Raperostelium*	Mushroom Research Center, Bahn Pa Dheng, Mae Taeng, Chiang Mai Province, Thailand	PQ834596
TH11CW	*Dictyostelium* sp. ^a^	1–192	*Dictyostelium* sp. *TH14B*	HQ141491.1	99.89%	*Raperostelium*	Doi Inthanon National Park, Chom Thong District, Chiang Mai Province, Thailand	PQ834594
TH14B	*Dictyostelium* sp. ^a^	1–195	*Dictyostelium* sp. *TH14B*	HQ141491.1	99.89%	*Raperostelium*	Doi Inthanon National Park, Chom Thong District, Chiang Mai Province, Thailand	PQ834595
Csub	*Cavenderia subdiscoidea* ^a^	2–103	*Dictyostelium* sp. *TH1A*	HQ141515.1 ^c^	99.95%	*Cavenderia*	Mushroom Research Center, Bahn Pa Dheng, Mae Taeng, Chiang Mai Province, Thailand	PQ834586
Cpse	*Cavenderia pseudoaureostipes* ^a^	1–68	*Dictyostelium* sp. *TH39A*	HQ141518.1 ^c^	99.98%	*Cavenderia*	Wat Pong AO (Temple), Mae Chan District, Chiang Rai Province, Thailand	PQ834587
TH18B	*Cavenderia ungulata* ^b^	1–188	*Dictyostelium* sp. *THC11X*	HQ141523.1 ^c^	100.00%	*Cavenderia*	Doi Inthanon National Park, Chom Thong District, Chiang Mai Province, Thailand	PQ834591
TH19B	*Cavenderia helicoidea* ^b^	1–110	*Dictyostelium* sp. *THC11X*	HQ141523.1 ^c^	100.00%	*Cavenderia*	Doi Inthanon National Park, Chom Thong District, Chiang Mai Province, Thailand	PQ834592
TH20A	*Cavenderia protumula* ^b^	1–111	*Dictyostelium* sp. *THC11X*	HQ141523.1 ^c^	100.00%	*Cavenderia*	Doi Inthanon National Park, Chom Thong District, Chiang Mai Province, Thailand	PQ834593
TH20C	*Cavenderia parvibrachiata* ^b^	1–39	*Dictyostelium* sp. *THC11X*	HQ141523.1 ^c^	99.73%	*Cavenderia*	Doi Inthanon National Park, Chom Thong District, Chiang Mai Province, Thailand	PQ834588
Cbhu	*Cavenderia bhumiboliana* ^a^	2–92	*Dictyostelium* sp. *THC11X*	HQ141523.1 ^c^	100.00%	*Cavenderia*	Doi Inthanon National Park, Chom Thong District, Chiang Mai Province, Thailand	PQ834590
Cpro	*Cavenderia protodigitata* ^a^	2–62	*Dictyostelium* sp. *THC11X*	HQ141523.1 ^c^	100.00%	*Cavenderia*	Doi Inthanon National Park, Chom Thong District, Chiang Mai Province, Thailand	PQ834589

Note: ^a^: The species name is as stated in the Dicty stock center (http://dictybase.org/StockCenter/StockCenter.html accessed on 18 April 2024). ^b^: The species name is as defined by the published literatures. ^c^: These accession numbers were mentioned in [8,12] as belonging to the genus *Cavenderia*. HQ141515.1, HQ141518.1 and HQ141523.1 were shown in the literature as the sequences of *C. subdiscoidea*, *C. pseudoaureostipes* and *C. bhumiboliana*, respectively.

**Table 2 microorganisms-13-00275-t002:** Pairwise differences of 16 cultured samples calculated from 18S rRNA gene sequences. The values above and below the diagonal line presented the proportion of pairwise difference sites and K2P distance, respectively. Proportions of pairwise differences and K2P distances equal to 0.00 were highlighted. “NA” highlighted in dark gray indicates it is not applicable.

	Dpur	Ddis	TH11C	TH18CC	TH14B (Delta)	TH8C	TH11CW	TH14B	Csub	Cpse	TH18B	TH19B	TH20A	Cbhu	Cprot	TH20C
*Dictyostelium purpureum*	NA	0.047	0.042	0.042	0.115	0.114	0.105	0.105	0.254	0.254	0.253	0.253	0.253	0.253	0.253	0.251
*Dictyostelium discoideum*	0.048	NA	0.034	0.034	0.110	0.111	0.113	0.113	0.257	0.255	0.254	0.254	0.254	0.254	0.254	0.254
*Dictyostelium* sp. *TH11C*	0.043	0.035	NA	0.000	0.111	0.111	0.111	0.111	0.258	0.257	0.257	0.257	0.257	0.257	0.257	0.255
*Dictyostelium* sp. *TH18CC*	0.043	0.035	0.000	NA	0.111	0.111	0.111	0.111	0.258	0.257	0.257	0.257	0.257	0.257	0.257	0.255
*Raperostelium* sp. *TH14B* (Delta)	0.126	0.119	0.121	0.121	NA	0.005	0.038	0.038	0.262	0.264	0.263	0.263	0.263	0.263	0.263	0.262
*Raperostelium* sp. *TH8C*	0.125	0.120	0.120	0.120	0.005	NA	0.041	0.041	0.261	0.265	0.263	0.263	0.263	0.263	0.263	0.262
*Raperostelium* sp. *TH11CW*	0.114	0.123	0.120	0.120	0.039	0.042	NA	0.000	0.262	0.263	0.264	0.264	0.264	0.264	0.264	0.263
*Raperostelium* sp. *TH14B*	0.114	0.123	0.120	0.120	0.039	0.042	0.000	NA	0.262	0.263	0.264	0.264	0.264	0.264	0.264	0.263
*Cavenderia subdiscoidea*	0.313	0.318	0.320	0.320	0.325	0.324	0.325	0.325	NA	0.009	0.006	0.006	0.006	0.006	0.006	0.007
*Cavenderia pseudoaureostipes*	0.313	0.316	0.318	0.318	0.328	0.330	0.327	0.327	0.009	NA	0.006	0.006	0.006	0.006	0.006	0.006
*Cavenderia ungulata*	0.312	0.313	0.318	0.318	0.326	0.326	0.328	0.328	0.006	0.006	NA	0.000	0.000	0.000	0.000	0.002
*Cavenderia helicoidea*	0.312	0.313	0.318	0.318	0.326	0.326	0.328	0.328	0.006	0.006	0.000	NA	0.000	0.000	0.000	0.002
*Cavenderia protumula*	0.312	0.313	0.318	0.318	0.326	0.326	0.328	0.328	0.006	0.006	0.000	0.000	NA	0.000	0.000	0.002
*Cavenderia bhumiboliana*	0.312	0.313	0.318	0.318	0.326	0.326	0.328	0.328	0.006	0.006	0.000	0.000	0.000	NA	0.000	0.002
*Cavenderia protodigitata*	0.312	0.313	0.318	0.318	0.326	0.326	0.328	0.328	0.006	0.006	0.000	0.000	0.000	0.000	NA	0.002
*Cavenderia parvibrachiata*	0.309	0.313	0.315	0.315	0.325	0.325	0.327	0.327	0.007	0.006	0.002	0.002	0.002	0.002	0.002	NA

**Table 3 microorganisms-13-00275-t003:** Summary of genetic relationship among samples previously defined as belonging to the same species. For ASAP and bPTP methods, MONO, MERGE, SPLIT, and multi-SPLIT were used for describing relationship among samples belonging to one species. MONO indicated that all samples of a species clustered into a single group. MERGE referred to cases where samples of a species were grouped with samples from other species. SPLIT was used when samples of a species were divided into multiple groups. And multi-SPLIT was used when samples were distributed across multiple groups, with each group carrying members of at least two species. For ML phylogeny, MONO and non-MONO refer to monophyletic and non-monophyletic relationships. # Samples and support (%) refer to number of samples and bootstrap supports. Success rate (%) and success rate with weight (%) were the success rate calculated based on the number of species and weighted by number of samples.

Species	# Samples	ASAP Analysis		ML Phylogeny		bPTP Analysis	
		Group Type	Group No.	Topology	Support (%)	Group Type	Group No.
*Dictyostelium ammophilum*	2	MERGE	6	MONO	98	MERGE	7
*Dictyostelium brefeldianum*	2	MERGE	5	non-MONO	N/A	MERGE	7
*Dictyostelium chordatum*	2	MONO	8	MONO	100	MERGE	7
*Dictyostelium clavatum*	2	MERGE	6	non-MONO	N/A	MERGE	7
*Dictyostelium dimigraformum*	2	MONO	4	MONO	99	MERGE	7
*Dictyostelium discoideum*	17	SPLIT	1, 2	non-MONO	N/A	MERGE	7
*Dictyostelium firmibasis*	4	MONO	3	MONO	100	MERGE	7
*Dictyostelium giganteum*	4	MONO	9	MONO	92	MERGE	7
*Dictyostelium leptosomopsis*	3	MERGE	5	non-MONO	N/A	MERGE	7
*Dictyostelium longosporum*	2	MERGE	6	non-MONO	N/A	MERGE	7
*Dictyostelium macrocephalum*	2	multi-SPLIT	6, 11	non-MONO	N/A	MERGE	7
*Dictyostelium mucoroides*	8	multi-SPLIT	5–7	non-MONO	N/A	MERGE	7
*Dictyostelium multiforme*	2	MERGE	6	non-MONO	N/A	MERGE	7
*Dictyostelium purpureum*	23	SPLIT	12–14	MONO	81	MERGE	7
*Dictyostelium septentrionalis*	2	MONO	10	MONO	93	MERGE	7
*Dictyostelium sphaerocephalum*	6	multi-SPLIT	6, 11	non-MONO	N/A	MERGE	7
*Polysphondylium patagonicum*	2	MERGE	15	non-MONO	N/A	MERGE	8
*Polysphondylium violaceum*	9	MERGE	15	non-MONO	N/A	MERGE	8
*Raperostelium minutum*	4	MONO	16	MONO	100	SPLIT	20, 24, 25
*Raperostelium tenue*	3	SPLIT	17–18	non-MONO	N/A	SPLIT	11, 12
*Coremiostelium polycephalum*	5	SPLIT	19–22	MONO	87	SPLIT	13, 21–23
*Synstelium polycarpum*	2	SPLIT	23–24	MONO	100	SPLIT	18, 19
*Rostrostelium ellipticum*	2	MONO	34	MONO	100	MONO	4
*Acytostelium amazonicum*	2	SPLIT	35, 36	MONO	100	SPLIT	5, 6
*Acytostelium leptosomum*	2	SPLIT	37, 38	MONO	99	SPLIT	2, 3
*Cavenderia aureostipes*	29	multi-SPLIT	39–42, 44, 45	non-MONO	N/A	MERGE	1
*Cavenderia bhumiboliana*	2	MERGE	45	non-MONO	N/A	MERGE	1
*Cavenderia* cf. *aureostipes*	19	multi-SPLIT	39–40, 43, 44–48	non-MONO	N/A	MERGE	1
*Cavenderia fasciculata*	8	multi-SPLIT	44, 48	non-MONO	N/A	MERGE	1
*Cavenderia helicoidea*	2	MERGE	45	non-MONO	N/A	MERGE	1
*Cavenderia parvibrachiata*	3	MERGE	45	MONO	88	MERGE	1
*Cavenderia protodigitata*	2	MERGE	45	non-MONO	N/A	MERGE	1
*Cavenderia protumula*	2	MERGE	45	non-MONO	N/A	MERGE	1
*Cavenderia pseudoaureostipes*	2	MERGE	45	MONO	100	MERGE	1
*Cavenderia subdiscoidea*	2	MERGE	42	MONO	98	MERGE	1
*Cavenderia ungulata*	2	MERGE	45	non-MONO	N/A	MERGE	1
*Heterostelium asymetricum*	2	MONO	29	MONO	100	MONO	17
*Heterostelium boreale*	2	MONO	30	MONO	100	SPLIT	26, 27
*Heterostelium candidum*	3	multi-SPLIT	25, 32	non-MONO	N/A	multi-SPLIT	10, 14
*Heterostelium colligatum*	3	MERGE	33	non-MONO	N/A	MERGE	15
*Heterostelium oculare*	2	MONO	31	MONO	100	MONO	9
*Heterostelium pallidum*	10	multi-SPLIT	25–26	non-MONO	N/A	multi-SPLIT	10, 16
*Heterostelium tenuissimum*	3	multi-SPLIT	26–28	non-MONO	N/A	MERGE	16
*Heterostelium tikalense*	2	MERGE	33	non-MONO	N/A	MERGE	15
Success rate (%)	44	22.73		43.18		6.82	
Success rate with weight (%)	214	12.15		31.31		2.80	

## Data Availability

The 18S rDNA sequences of 16 cultured samples presented in the study are available in GenBank database under accession PQ834586-601.

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
