# Peer review of "Limitations of 18S rDNA Sequence in Species-Level Classification of Dictyostelids"

_microorganisms, 2025, doi:10.3390/microorganisms13020275_

Round 1
Reviewer 1 Report
Comments and Suggestions for Authors
Comments
The manuscript presents an interesting study examining the limitations of using the 18S rDNA sequence for species-level classification in dictyostelids, a group of soil-dwelling amoebae known for their complex life cycles and morphological diversity. The authors conduct a thorough analysis, employing various methods including sequence data collection, phylogenetic analysis, and species delimitation techniques. The results provide valuable insights into the strengths and weaknesses of using 18S rDNA for taxonomic classification in this group of organisms.
Strengths:
Comprehensive Analysis: The authors conduct a comprehensive analysis by collecting and comparing 18S rDNA sequences from multiple dictyostelid species. They employ various bioinformatics tools and methods to assess the variability within and between species.
Methodological Rigor: The methodology employed in the study is rigorous and well-documented. The authors describe in detail the steps taken for sample collection, sequence amplification, data analysis, and species delimitation.
Useful Results: The results obtained provide useful insights into the limitations of using 18S rDNA for species-level classification in dictyostelids. The authors highlight the variability within species and the challenges in delineating species boundaries based solely on 18S rDNA sequences.
Weaknesses and Suggestions for Improvement:
Clarity in Introduction: While the introduction provides a good overview of dictyostelids and their life cycles, it could benefit from a clearer statement of the research question and hypothesis. Specifically, the authors should state more explicitly what they aim to investigate and what they expect to find regarding the limitations of 18S rDNA for species-level classification.
Discussion of Results: The discussion section could be more in-depth, particularly in interpreting the results and drawing conclusions. The authors should provide a more nuanced discussion of the reasons behind the observed limitations of 18S rDNA, including potential sources of error and the impact of sample size on the results.
Specific Comments:
Line 109:
Please provide more details about the sample collection process, including the number of samples collected, the geographical distribution of the samples, and any criteria used for sample selection.
Line 123:
The authors should clarify how the sequence data were collected and curated, including any quality control measures taken to ensure the accuracy of the sequences.
Line 191:
The authors should provide more information about the phylogenetic analysis, including the software and algorithms used, the number of bootstrap replicates, and any assumptions or limitations of the method.
Line 315:
The authors should provide more detail about the barcoding gap analysis, including how the barcoding gaps were calculated and the significance of the results.
Line 366:
The authors should discuss the reasons for the low success rate of species delimitation using ASAP and suggest potential improvements or alternative methods.
In conclusion, the manuscript presents an interesting and important study on the limitations of using 18S rDNA for species-level classification in dictyostelids. With some minor revisions and improvements, the manuscript has the potential to make a significant contribution to the field of taxonomy and biodiversity research.
Reviewer 2 Report
Comments and Suggestions for Authors
This manuscript concerns the taxonomy of cellular slime molds, the dictyostelids. As in many other groups of microorganisms, also in dictyostelids the traditional morphology-based taxonomy is being increasingly replaced by molecular phylogeny. The main aim of the present work was to determine the strengths and limitations of the 18S rDNA sequence for species-level classification. Detailed studies and analyses were performed using sixteen dictyostelid samples. The sequences of 150 species belonging to 12 genera were collected from the GenBank database. The results of species delimitation analysis using the ASAP method, the maximum likelihood (ML) phylogeny and Bayesian implementation of the Poisson tree process model (bPTP) are presented. The usefulness of these methods is assessed. The results of the analyses performed are presented in a clear and transparent manner. The most important result of the work is the confirmation of the effectiveness of 18S rDNA sequence for genus-level classification and demonstration of its limitations for species-level classification in dictyostelids. The authors indicated many factors that influenced this limitation. It would be advisable for the authors to compare in Discussion the results obtained for dictyostelids with how this situation presents itself in Fungi. The manuscript is carefully prepared. Only a few typographical errors were noted, which are listed in Remarks.
Line 79 et al.[11] - add a space
Line 122 typographical errors
Line 169 this is unclear "Dictyostelium (A–D, A–P), it should be (Dictyostelium (A–D, J–P) ??
Line 169 Could the photos in Figure 1 be clarified a bit?
Line 237-240 "… sp." should be written not italic (this also applies to other places in the entire manuscript)
Line 267 in Table 1 and Line 312 in Table 2 should be made numerous corrections - sp should be written as sp. (with a dot) and not italic
Line 372, line 410 and in Table 3 cf. - it should not be italic
Line 472 Text in Figure 4 is unreadable
